

# Insights into mobile genetic elements and the role of conjugative plasmid in transferring aminoglycoside resistance in extensively drug-resistant *Acinetobacter baumannii* AB329

Supat Khongfak[1], Rapee Thummeepak[1], Udomluk Leungtongkam[1], Kannipa Tasanapak[1], Aunchalee Thanwisai[1] and Sutthirat Sitthisak[1,2]

[1] Department of Microbiology and Parasitology, Faculty of Medical Science, Naresuan University, Muang, Phitsanulok, Thailand
[2] Centre of Excellence in Medical Biotechnology, Faculty of Medical Science, Naresuan University, Muang, Phitsanulok, Thailand

Corresponding author
Sutthirat Sitthisak,
Sutthirats@nu.ac.th

## ABSTRACT

*Acinetobacter baumannii* is a major cause of nosocomial infection, and the incidence of extensively drug-resistant *A. baumannii* (XDRAB) infections has dramatically increased worldwide. In this study, we aimed to explore the complete genome sequence of XDRAB 329, ST1166/98 (Oxford/Pasteur), which is an outbreak clone from a hospital in Thailand. Whole-genome sequencing (WGS) was performed using short-read Illumina and long-read PacBio sequencing, and a conjugation assay of its plasmid was performed. The complete genome sequence of *A. baumannii* AB329 revealed a circular chromosome 3,948,038 bp in length with 39% GC content. Antibiotic resistance genes (ARGs), including beta-lactam resistance ($bla_{OXA-51}$, $bla_{ADC-25}$, $bla_{OXA-23}$, $bla_{TEM-1D}$), aminoglycoside resistance ($aph(3')$-Ia, $aph(3'')$-Ib, $aph(6)$-Id, $arm$A), tetracycline resistance ($tet$(B), $tet$ (R)), macrolide resistance ($mph$(E), $msr$(E)), and efflux pumps, were found. Mobile genetic elements (MGEs) analysis of *A. baumannii* AB329 revealed two plasmids (pAB329a and pAB329b), three prophages, 19 genomic islands (GIs), and 33 insertion sequences (ISs). pAB329a is a small circular plasmid of 8,731 bp, and pAB329b is a megaplasmid of 82,120 bp. $aph(3')$-VIa was detected in pAB329b, and a major facilitator superfamily (MFS) transporter was detected in the prophage. *Acinetobacter baumannii* resistance island 4 (AbaR4) harboring tetracycline and aminoglycoside resistance was detected in the genome of *A. baumannii* AB329. pAB329b, which belongs to Rep-type GR6 (plasmid lineage LN_1), is a conjugative plasmid with the ability to transfer an aminoglycoside resistance gene to sodium azide-resistant *A. baumannii*. This study provides insights into the features of the MGEs of XDRAB, which are the main reservoir and source of dissemination of ARGs.

## INTRODUCTION

*Acinetobacter baumannii* is a bacterium that is a major cause of nosocomial infection, especially in intensive care units (ICUs). In recent decades, the prevalence of extensively drug-resistant *A. baumannii* (XDRAB) has been rapidly increasing worldwide. Numerous antibiotic resistance genes (ARGs) have been detected in the genomes and mobile genetic elements (MGEs) of XDRAB, and they have been found to be responsible for the spread of antibiotic resistance. A variety of MGEs have been described in *A. baumannii*. *In silico* analysis detected various ARGs located in *A. baumannii* conjugative plasmids (*Makke et al., 2020*; *Martins-Sorenson et al., 2020*). A conjugative plasmid is a self-replicating plasmid that generally carries all the genes required for bacterial conjugation. The conjugative plasmid requires the origin of transfer and the *tra* operon, which is important for generating the F-pilus that is needed for transferring genetic materials. Plasmid classification of *A. baumannii* can be divided into 23 different groups (GR1- GR23) based on the nucleotide identity of the replicase genes (*repA*) using the polymerase chain reaction-based replicon typing (AB-PBRT) method (*Bertini et al., 2010*; *Salgado-Camargo et al., 2020*). *Kongthai et al. (2021)* reported the detection of *tra* genes in XDRAB strains harbored in plasmid GR6. This GR6 plasmid is responsible for the dissemination of drug resistance genes such as $bla_{OXA-23}$ and *aphA6* (*Saranathan et al., 2014*; *Leungtongkam et al., 2018a*). Additionally, *A. baumannii* plasmids were also classified into 21 lineages (LN_1- LN_21) based on their core DNA sequence backbones and plasmid incompatibility groups (*Salgado-Camargo et al., 2020*).

Prophages, which are bacteriophage genomes integrated into the bacterial genome, are unique MGEs that constitute 10–20% of the host genome and provide new genetic information, such as virulence factors and drug resistance mechanisms (*Casjens, 2003*). A previous study by *Loh et al. (2020)* investigated 177 *A. baumannii* genomes and determined that the number of prophages ranged from one to 15 regions, and less than 5% of the genomes contained prophage-encoded ARGs (*Loh et al., 2020*). Other MGEs, such as integrons (In), transposons (Tn), and insertion sequences (ISs), were also found to be related to antibiotic resistance in *A. baumannii*. Integrons are DNA elements consisting of the integrase gene (*Int*), the integron-associated recombination site, and gene cassettes carrying ARGs (*Gillings, 2014*). Insertion sequences (ISs) are small transposable elements (transposes genes) that have the ability to move within the bacterial genome, and transposons (Tn) are gene cassettes consisting of two inverted repeats from two separate transposons moving together as one unit and carrying the ARGs between them (*Nigro & Hall, 2015*). The predominant carbapenem resistance gene in *A. baumannii*, $bla_{OXA-23}$, was found to be associated with IS*Aba1*, IS*Aba4*, IS*Aba10*, Tn2006, Tn2007, Tn2008, Tn2008b, and Tn2009 (*Nigro & Hall, 2015*; *Hamidian & Nigro, 2019*). The other MGEs with their associated ARGs are IS*Aba1* ($bla_{OXA-5}$, $bla_{OXA-58}$, $bla_{AmpC}$), IS*Aba2* ($bla_{OXA-58}$, $bla_{AmpC}$), IS*Aba3* ($bla_{OXA-58}$), IS*Aba125* ($bla_{NDM-1}$, $bla_{NDM-2}$, $bla_{AmpC}$, *aphA6*), IS18 ($bla_{OXA-58}$), *Int1* ( $bla_{GES-11}$, $bla_{GES-14}$, *dfrA1*, *sat2*, *aadA1*, *orfX*, *ybfA*, *ybfB*), and *Int2* (*dfrA1*, *sat2*, *aadA1*, *orfX*, *ybfA*, *ybfB*, *ybgA*) (*Pagano, Martins & Barth, 2016*; *Turton et al., 2006*; *Frontiers Production Office, 2015*; *Joshi et al., 2017*). In addition, different ISs located

upstream and/or downstream of ARGs increase the transcription of ARGs. Genomic islands (GIs) are regions of bacterial genomes that are acquired by horizontal gene transfer (HGT). AbaR-type genomic islands (AbaRs) are important elements responsible for antimicrobial resistance in *A. baumannii*. Several AbaRs have been characterized; the majority, such as AbaR1, AbaR3, AbaR5, AbaR6, AbaR7, AbaR8, AbaR9, and AbaR10, were identified in epidemic clones such as international clone (IC) one (*Pagano, Martins & Barth, 2016*). Others were identified in IC2, such as AbaR4 (*Kim, Park & Ko, 2012*).

The two major clones responsible for *A. baumannii* outbreaks worldwide are global clone 1 (GC1) and global clone 2 (GC2) but are referred to as international clones 1 (IC1) and 2 (IC2) (*Hamidian & Nigro, 2019*). The global distribution of CRAB has been heavily influenced by the spread of IC2 isolates, and most of these clones are defined as sequence type 2 (ST2) according to the Institut Pasteur MLST scheme (*Hamidian & Nigro, 2019*). IC2 is the predominant clonal lineage in Spain, South Korea, China, Australia, Singapore, and Thailand (*Hamidian & Nigro, 2019*, *Khuntayaporn et al., 2021*). Our previous study of 339 *A. baumannii* isolates collected from four hospitals in Thailand revealed 7.9% XDRAB among the total isolates collected (*Leungtongkam et al., 2018a*). We found an outbreak clone of XDRAB in one hospital with the same ST type and plasmid group that belonged to IC2 (*Kongthai et al., 2021*). All of them were ST98 (Pasteur) and contained GR2 and GR6 plasmids. Little is known about the role of conjugative plasmids in the functioning of XDRAB. The use of whole-genome sequencing (WGS) technology can assist in tackling antimicrobial resistance, virulence determinants, and MGEs in *A. baumannii* (*Makke et al., 2020*; *Martins-Sorenson et al., 2020*). Combined with short- and long-read sequencing, WGS will be able to resolve an accurate and complete genome and plasmid structure. Thus, this study aimed to obtain the complete genome sequence of XDRAB and to characterize the MGEs and the role of the plasmid in transferring ARGs.

## MATERIAL AND METHODS

### Bacterial strains and antibiotic susceptibility testing (AST)

*A. baumannii* AB329, which is a phage-susceptible XDRAB strain, was isolated from patient sputum obtained from our previous study (*Leungtongkam et al., 2018a*). This strain is representative of the outbreak clone obtained from a hospital in east Thailand, and it was collected from November 2013 to February 2015 (*Leungtongkam et al., 2018a*). Antimicrobial susceptibility testing (AST) was performed as previously described (*Kongthai et al., 2021*). The AST results were interpreted according to the Clinical Laboratory Standard Institute (*CLSI, 2020*). The protocol was approved by the Naresuan University Institutional Biosafety Committee, and the project number was NUIBC MI 63-07-21.

### Polymerase chain reaction (PCR)-based replicon typing method

The AB-PBRT method was used to detect plasmid groups (GRs) with primers specific to each GR from GR1 to GR19, which were described by a previous study (*Bertini et al., 2010*).

### DNA extraction, genome sequencing and assembly

Genomic DNA of *A. baumannii* AB329 was extracted using the Real Genomics DNA Extraction Kit (RBC Biosciences, Taiwan), and it was quantified using a Qubit® DNA

Assay Kit in a Qubit® 2.0 Fluorometer (Life Technologies, CA, USA) prior to sequencing with both short-read (Illumina paired-end) and long-read (PacBio, Menlo Park, CA, USA) sequencing systems. For short-read sequencing, paired-end sequencing libraries (2× 250 bp) were constructed using the Nextera XT sample preparation kit following the manufacturer's suggestions, and they were sequenced using the Illumina MiSeq platform. The Illumina reads were trimmed with Sickle v1.33 using the default parameters (*Joshi & Fass, 2011*). For long-read sequencing, a large insert library (10 kb) was constructed and sequenced on the PacBio RS platform (Pacific Biosciences, Menlo Park, CA, USA). A hybrid assembly was conducted with the Illumina trimmed reads and PacBio reads using Unicycler v0.4.8.0 with the default settings (*Wick et al., 2017*). Unicycler automatically identified and trimmed the overlapping ends, and circular sequences were rotated to *dnaA*.

## Genome annotation and bioinformatic analysis

The assembled circular chromosome and plasmids were functionally annotated using Prokka v1.12 with the default options. MLST types were determined *in silico* using the MLST database (https://pubmlst.org/organisms/acinetobacter-baumannii). The complete genomes of 292 *A. baumannii* strains retrieved from the NCBI database in December 2021 were used to identify the core genome (Table S1). A single-nucleotide polymorphism (SNPs) phylogenetic tree of the core genome was reconstructed using CSI Phylogeny with the default settings (*Kaas et al., 2014*). This tree was visualized and edited using Interactive Tree of Life (iTOL) (https://itol.embl.de/). A pangenomic analysis was executed using Roary v.3.13.0, which compared the five closest related genomes, including *A. baumannii* NIPH17_00019 (AP024415.1), *A. baumannii* XH856 (CP014541.1), *A. baumannii* KAB02 (CP017644.1), *A. baumannii* KAB06 (CP017652.1) and *A. baumannii* KAB05 (CP017650.1). Then, the output was illustrated using R studio as described in https://github.com/IamIamI/pADAP_project/tree/master/Roary_stats. The average nucleotide identity (ANI) was calculated using FastANI v.1.3 to estimate the whole-genome similarity among the two XDRAB strains, which were obtained from hospitalized patients in Thailand (*Kongthai et al., 2021*), and the five closest related genomes identified from the pan-genome analysis. Antimicrobial resistance and virulence genes were retrieved using the Comprehensive Antibiotic Resistance Database (CARD) and VFanalyzer (*Liu et al., 2019*), respectively. The large-scale BLAST score ratio (LS-BSR) pipeline was utilized to compare the virulence and drug resistance genes with 292 *A. baumannii* genomes (Table S1). A BSR value of 0.4 and above was interpreted as the presence of genes, and a BSR value below 0.4 was inferred as gene absence (*Sahl et al., 2014*; *Yakkala et al., 2019*). Then, these BSR values were used to build a hierarchical clustering heatmap using the R packages pheatmap and tidyverse. MGEs were detected using MobileElementFinder (*Johansson et al., 2021*). The presence of prophage sequences in the genome of *A. baumannii* AB329 was analyzed using the PHAge Search Tool Enhanced Release (PHASTER) online server (*Arndt et al., 2016*). Prophage open reading frames (ORFs) were examined as described in a previous study (*Fu et al., 2017*). The prediction of genomic islands (GIs) and AbaR was conducted by running BLASTn. A set of nucleotide queries of GIs identified in *A. baumannii* ACICU was used for BLASTn searching (*Di Nocera et al., 2011*; *Thummeepak et al., 2020*).

Plasmid comparison and the identification of the AbaR structure were accomplished using Easyfig version 2.1 (*Sullivan, Petty & Beatson, 2011*). The complete genome was deposited in the NCBI GenBank database under the accession numbers CP091452 (chromosome), CP091453 (pAB329a), and CP091454 (pAB329b).

### Conjugation experiment

A broth-mating conjugation assay was performed according to a previously published protocol with minor modifications (*Leungtongkam et al., 2018b*). Overnight cultures of the donor (*A. baumannii* AB329) and the sodium azide resistance recipient (*A. baumannii* NU13R) were adjusted in 0.85% NaCl until the cell suspensions reached a turbidity equal to a McFarland value of 0.5, which was measured using a densitometer (SiaBiosan, Riga, Latvia). Equal volumes (250 µl) of adjusted cell suspensions of the donor and the recipient were mixed in 500 µl of 2× Luria-Bertani (LB) broth and incubated for 4 h at 37 °C. Transconjugants were selected on three LB agar plates containing 250 µg/ml sodium azide (negative control), 20 µg/ml kanamycin (negative control), and 250 µg/ml sodium azide plus 20 µg/ml kanamycin.

The conjugation frequency (CF) was calculated as previously described (*Leungtongkam et al., 2018b*). PCR to detect plasmid groups (*Bertini et al., 2010*), aminoglycoside resistance genes (*Kongthai et al., 2021*) and *traU* genes (*Kongthai et al., 2021*) was performed using cell lysates from donor (AB329), recipient (NU13R), and transconjugants (NU13R-pAB329b) as templates.

## RESULTS

### Antibiotic susceptibility testing (AST)

Antibiotic susceptibility testing of *A. baumannii* AB329 using the disk diffusion method revealed that it was resistant to imipenem, meropenem, amikacin, ciprofloxacin, gentamicin, trimethoprim/sulfamethoxazole, cefoperazone/sulbactam, colistin, and tigecycline (Table S2). The minimum inhibitory concentrations (MICs) of imipenem, colistin, and tigecycline were 32, 1 and 0.25 ug/mL, respectively (Table S2).

### Complete genome sequence of *A. baumannii* AB329

The complete genome sequence of *A. baumannii* AB329 generated by short- and long-read sequencing revealed a circular chromosome 3,948,038 bp in length with 39% GC content, and it contained two plasmids (pAB329a and pAB329b) (Table 1). pAB329a is a small circular plasmid of 8,731 bp, and pAB329b is a megaplasmid of 82,120 bp. The Prokka prokaryotic genome annotation system identified 18 rRNAs, 72 tRNAs, 3,837 ORFs, and a total of 3,747 protein-coding genes on the main chromosome of AB329 (Table S3). Genes for *tRNAs* and *rRNAs* were detected only in the chromosome (Table 1). pAB329a contained 12 ORFs, while pAB329b contained 113 ORFs. AB329 was assigned to MLST type 1166/98 (Oxford/Pasteur) and was found to belong to the IC2 lineage.

**Table 1  Genome features of the extensively drug-resistant *Acinetobacter baumannii* AB329.**

| Genome characteristics | AB329 (chromosome) | pAB329a (plasmid1) | pAB329b (plasmid2) |
|---|---|---|---|
| **General features** | | | |
| Genome size (bp) | 3,948,038 | 8,731 | 82,120 |
| Topology | circular | circular | circular |
| GC content (%) | 39.0 | 34.4 | 33.7 |
| Number of ORFs | 3,837 | 12 | 113 |
| Number of CDSs | 3,747 | 12 | 113 |
| Number of tRNAs | 72 | nd | nd |
| Number of rRNAs | 18 | nd | nd |
| *In silico* MLST (Pasteur/Oxford) | 98/1166 | nd | nd |
| **Insertion Sequences (ISs)** | | | |
| Number of total ISs | 34 | nd | 3 |
| Number of ISAba1 | 17 | nd | 1 |
| Number of ISAba13 | 1 | nd | nd |
| Number of ISAba24 (IS66) | 1 | nd | nd |
| Number of ISAba26 | 5 | nd | nd |
| Number of ISAba33 | 9 | nd | nd |
| Number of IS26(IS6) | 1 | nd | nd |
| Number of ISAba125 | nd | nd | 2 |
| **Number of total prophage regions** | | | |
| Intact prophage | 1 | nd | nd |
| Incomplete prophage | 2 | nd | nd |
| **Genomic islands** | | | |
| Number of total GIs | 19 | nd | nd |

**Notes.**
nd, gene or DNA element was not detected.

## Phylogenomic and comparative genomic analyses of *A. baumannii* AB329

Phylogenomic analysis was performed using the core genome of *A. baumannii* AB329 and 292 additional *A. baumannii* strains deposited in the NCBI database. As shown in Fig. 1A, phylogenetic analysis of *A. baumannii* AB329 presented in the same cluster with the *A. baumannii* strains NIPH17_00019 (AP024415.1), XH856 (CP014541.1), KAB02 (CP017644.1), KAB06 (CP017652.1), and KAB05 (CP017650.1) (Fig. 1B and Table S4). We also compared the genome of *A. baumannii* AB329 with two XDRAB isolates from two different hospitals in Thailand as previously described (*Kongthai et al., 2021*). The ANI (%) values of *A. baumannii* AB329 with *A. baumannii* AB140 and *A. baumannii* AB053 were 99.72% and 99.27%, respectively (Table S4). A pangenome of *A. baumannii* AB329 consisting of 4,628 genes represented the core, shell, and cloud genomes (Fig. 1B). The core genome represented a pool of conserved genes that were present in all genomes and included 3,238 genes. The accessory genes, which included shell genes (genes present in two or more strains) and cloud genes (genes only found in a single strain), constituted a total of 1,390 genes.

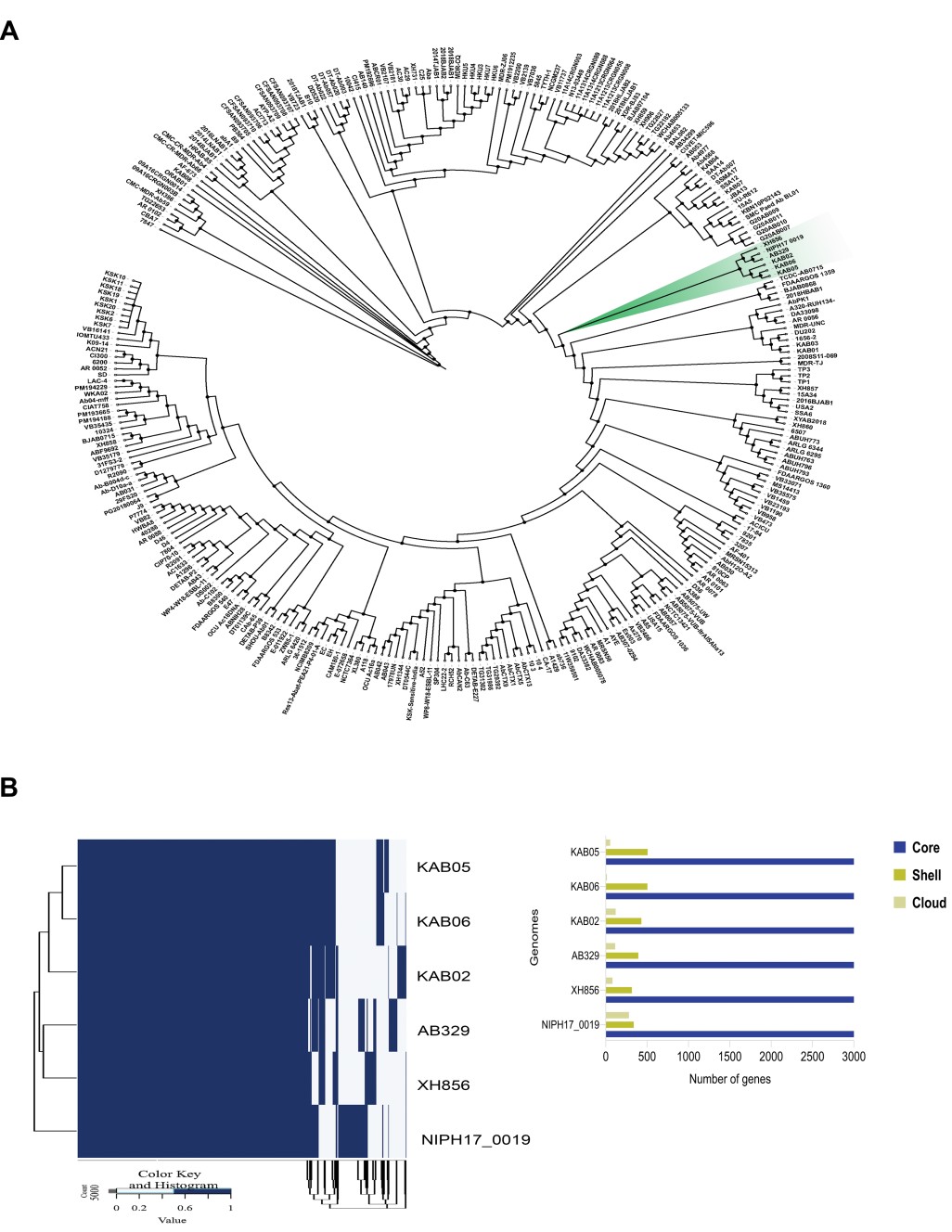

**Figure 1** Phylogenomic tree based on core-genome SNPs of *A. baumannii* AB329 and 292 *A. baumannii* genomes deposited in the NCBI database (A) and comparative genomic analysis of the pangenome identified in *A. baumannii* AB329 and its closely related genomes (B).

## Virulence genes, antibiotic resistance genes, and mobile genetic elements of *A. baumannii* AB329

Analysis of the virulence genes of *A. baumannii* AB329 revealed that genes were involved in iron uptake (*hemO, barA, barB, basA, basB, basC, basD, basF, basG, bash, basI, basJ,*

*bauA, bauB, bauC, bauD, bauE, bauF, entE*), serum resistance (*pbpG*), stress adaptation (*katG, katE*), gene regulation (*bfmR, bfmS*), immune evasion (*lpsB, lpxA, lpxB, lpxC, lpxD, lpxL, lpxM*), enzyme phospholipase (*plcC, plcD*), biofilm formation (*adeF, adeG, adeH, csuA, csuB, csuC, csuD, csuE, pgaA, pgaB, pgaC, pgaD*), and host cell adherence (*ompA*) (Fig. 2A). Resistome analysis detected a number of resistance mechanisms in the chromosome, including beta-lactam-inactivating enzymes, aminoglycoside-modifying enzymes, efflux pumps, permeability defects, and target site modifications. As shown in Fig. 2B, genes conferring drug resistance, including beta-lactam resistance (*bla*$_{OXA-51}$, *bla*$_{ADC-25}$, *bla*$_{OXA-23}$, *bla*$_{TEM-1D)}$), aminoglycoside resistance (*aph(3′)*-Ia, *aph* (3″)-Ib, *aph* (6)-Id, *arm* A), tetracycline resistance (*tet* (B), *tet* (R)), and macrolide resistance (*mph* (E), *msr* (E)), were detected on the chromosome. Genes encoding resistance-nodulation-cell division (RND) (*Ade* ABC, *Ade* IJK, and *Ade* FGH), multidrug and toxic efflux (MATE) (*Abe* M), major facilitator superfamily (MFS) (*tet* (A) and *tet* (B)), and small multidrug resistance (SMR) efflux systems (*Abe* S) were found. *In silico* analysis of the pattern of ARGs was conducted, and the 292 selected *A. baumannii* genomes worldwide were grouped into three clusters (A, B, and C) (Fig. 2B). We found that *A. baumannii* AB329 was grouped into Cluster C and was closely related to *A. baumannii* VB2181 (CP050401.1) and AC29 (CP007535.2), which were isolated from India and Malaysia. The MGEs of *A. baumannii* AB329 included two plasmids, three prophages, 19 GIs, and 33 ISs. The two plasmids were pAB329a and pAB329b. pAB329a is a small circular plasmid of 8,731 bp, and pAB329b is a megaplasmid of 82,120 bp. pAB329b carries conjugation-gene clusters required for autonomous conjugative transfer, which involves F pilus (*traD, traE, traK, traB, traV, traC, traW, traN, traF, traH, traG, traW*), plasmid replication (*repA*), and a recombinase (*recD2*). A few hypothetical proteins and one copy of the aminoglycoside resistance gene *aph* (*3′*)-VIa were detected in pAB329b (Fig. 3A). The genomic features of plasmid pAB329b aligned with 11 closely related plasmids deposited in the GenBank database are represented in Fig. 3A. pAB329a was closely related to plasmid p2AB5075 (CP008708.1) in lineage_2(LN_2), and pAB329b was similar to pACICU2 (NC_010606) in LN-1 (Table S5). Compared to pACICU2, we found that both plasmids had the same backbone regions; however, pAB329b harbored the *aph(3′)*- VIa gene, while it the ARG was absent from pACICU2 (Fig. 3B). We found three prophages, one intact and two incomplete, in the genome of *A. baumannii* AB329. Our bioinformatic analysis revealed that the intact prophage sequence contained 69 ORFs involved in DNA processing, drug resistance, host lysis, integrase, metabolism process, and phage proteins (Fig. 4A). Interestingly, the MFS transporter was detected in the genome of this prophage, which was homologous to *Acinetobacter* phage YMC11/11/R3177 (KP861230.1), with 57.98% ANI. The genome of *A. baumannii* AB329 was examined for GIs, and 19 were identified (Table S6). In addition, one resistance island, AbaR4, which harbored tetracycline and aminoglycoside resistance genes, was detected in the genome of *A. baumannii* AB329 (Fig. 4B). Transposable elements such as transposons and ISs were investigated, and 34 ISs were detected in the chromosome, except for *ISAba* 1, which was detected in the chromosome and the plasmid (Table 1). *ISAba*125 was detected only in the plasmid, and it bracketed the *aph* (3′)-VIa gene (Fig.

**Table 2  Conjugal transfer of the plasmid pAB329b and its contribution to aminoglycoside resistance.**

| Characteristics | AB329 | NU13R | NU13R-pAB329b |
|---|---|---|---|
| Conjugation frequency (CF) | – | – | $1.2 \times 10^{-7}$ |
| Antibiogram of Kanamycin (Disc diffusion) | Resistant | Susceptible | Resistant |
| MIC of kanamycin ($\mu$g/ml) | >64 $\mu$g/ml | 2 $\mu$g/ml | >64 $\mu$g/ml |
| PCR-based plasmid group typing | GR2, GR6 | absent | GR6 |
| Aminoglycoside resistance genes | *aph(3′)*-VIa | absent | *aph(3′)*-VIa |
| *traU* (gene in *tra* operon) | present | absent | present |

Notes.
PCR was performed to detect plasmid group (*repA*), aminoglycoside resistance gene (*aph(3′)*-VIa), and *TraU* gene.

3A). The ARGs in the chromosome located near the ISs were $bla_{OXA-23}$ (ISAba1), $bla_{TEM}$ (ISAba33), and *aph(3′)*-VIa ((IS6).

## Conjugative transfer of the aminoglycoside resistance gene of plasmid pAB329b

To investigate the role of pAB329b in the transfer of ARG, we performed a conjugation assay to study the plasmid's ability to transfer the aminoglycoside resistance gene to sodium azide-resistant *A. baumannii* NU13R (recipient). As shown in Table 2, aminoglycoside resistance could be transferred from the donors (*A. baumannii* AB329) to the recipient (*A. baumannii* NU13R). The conjugation frequency was approximately $1.2 \times 10^{-7}$. The resistance genes, plasmid typing results, and antibiotic susceptibility of the donors, the recipient, and the transconjugants are shown in Table 2. The transconjugant MIC of kanamycin was > 64 $\mu$g/ml, which is approximately 32-fold higher than the MIC of kanamycin on the recipient strains. PCR amplification of aminoglycoside resistance genes (*aph(3′)*- VIa) and *traU* revealed their presence only in donors and transconjugants (Table 2). The plasmid GR6 was detected in the transconjugant NU13R-pAB329b.

## DISCUSSION

The incidence of XDRAB infection has increased, resulting in hospital outbreaks worldwide, including in Thailand. In this study, we investigated the genome features and MGEs of the XDRAB strain AB329. The complete genome sequence of *A. baumannii* AB329 was 3.9 Mb compared to that of previously reported XDRAB isolates, which ranged from 3.8 to 4.0 Mb (*Chopjitt et al., 2020*; *Si-Tuan et al., 2020*; *Makke et al., 2020*). The most dominant sequence type of multidrug-resistant *A. baumannii* (MDRAB) in Thailand was found to be ST2 (Pasteur), belonging to IC2 (*Khuntayaporn et al., 2021*; *Chukamnerd et al., 2022*), and the XDRAB ST types reported in Thailand were ST2, ST16, and ST1479 (*Chopjitt et al., 2020*; *Kongthai et al., 2021*). We found that the ST type of the *A. baumannii* AB329 strain was ST98 (Pasteur). ST98 is closely related to ST2 since only the *cpn60* locus from seven loci of MLST typing is different. To date, the ST98 clone has been detected in the carbapenem-resistant *A. baumannii* (CRAB) strain isolated from Portugal (*Silva et al., 2021*). An analysis of the clonal relationship of *A. baumannii* AB329 among 292 *A. baumannii* isolates worldwide (Table S2) revealed the highest genome similarity with the *A. baumannii* strains KAB02 (CP017644.1), KAB05 (CP017650.1), KAB06 (CP017652.1),

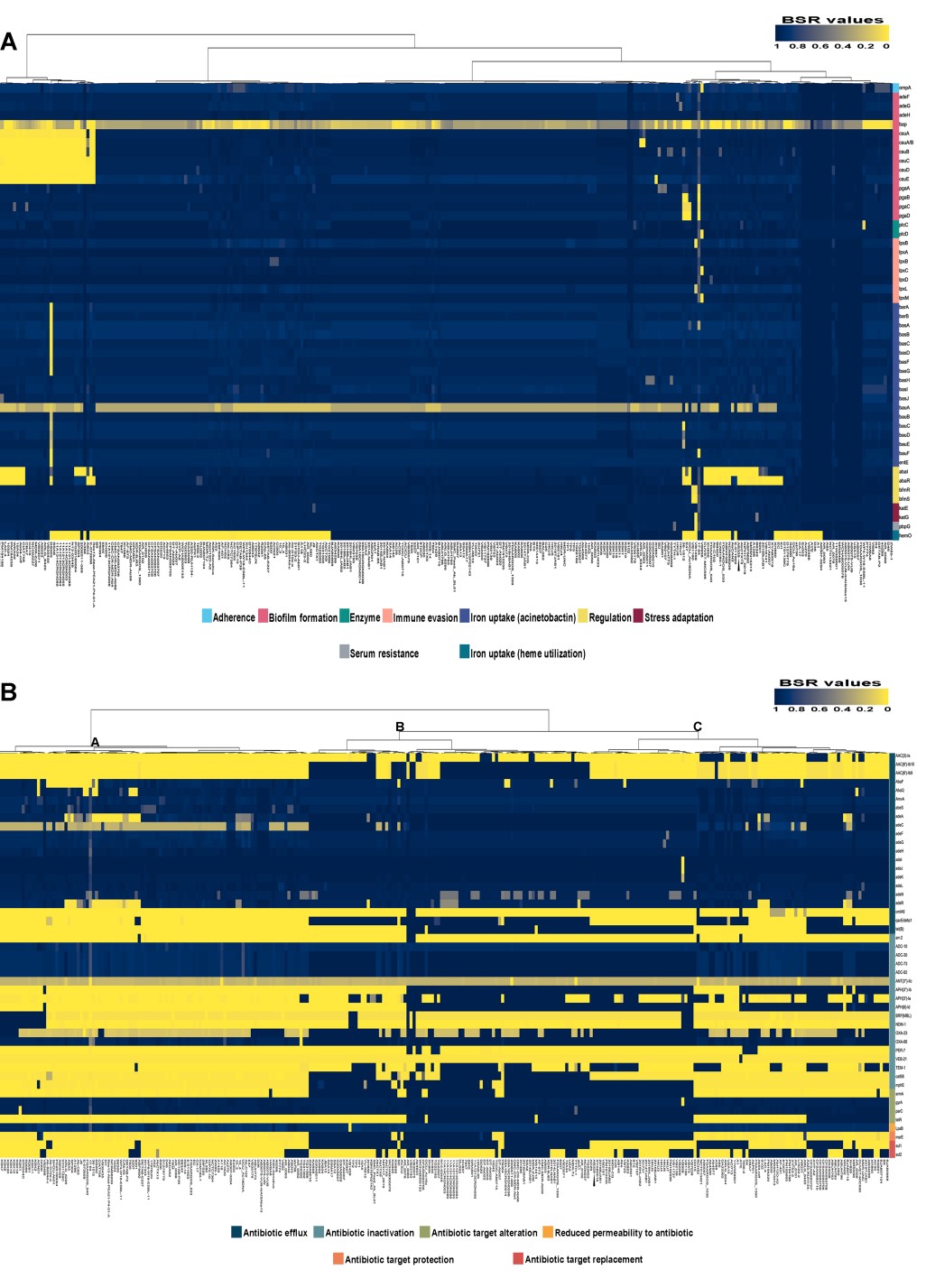

**Figure 2  Heatmap and antibiotic resistome of *A. baumannii* AB329.**  Heatmap showing LS-BSR analysis of the virulome (A) and antibiotic resistome (B). *A. baumannii* AB329 is marked with a black triangle for virulome analysis while showing antibiotic resistome analysis in Cluster C.

**A**

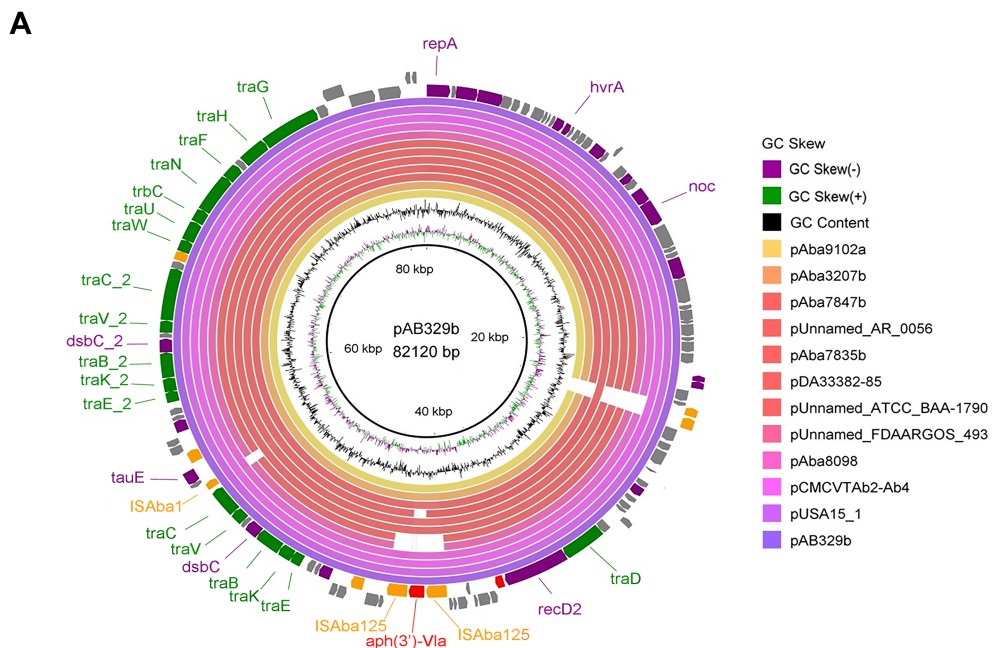

**B**

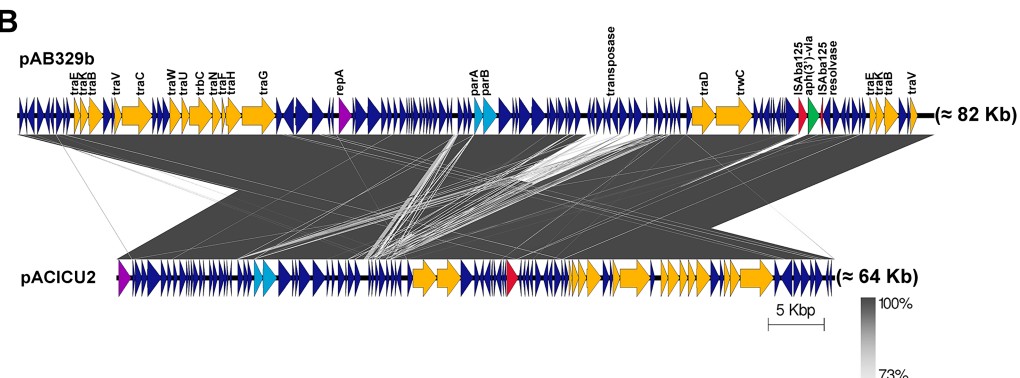

**Figure 3** **Map of the conjugative plasmid pAB329b and backbone comparison between all regions of the plasmids pAB329b and pACICU2.** Circular map of the GR6 conjugative plasmid pAB329b and multiple plasmid comparisons with its 11 closest relatives deposited in the GenBank database (A). The outer circle represents the ORFs, and their orientations are color-coded by functional category: navy: conserved hypothetical, green: Type IV secretion system (conjugation), red: drug- or putative virulence-associated proteins, orange: intact IS or transposase, and purple: plasmid replication, maintenance, or other functions. Backbone comparisons were made between all regions of the plasmids pAB329b and pACICU2 (LN_1) (B). Arrows represent the identified ORFs and are oriented in accordance with their direction. Homologous regions are highlighted in dark gray, while the backbone regions are shown using yellow arrows.

XH856 (CP014541.1), and NIPH17_00019 (AP024415.1) isolated from South Korea, China and Cambodia (Table S4). These bacterial strains might share a common ancestor since all six strains shared conserved homologous core genes (approximately 3,300 genes) (*Harris et al., 2003*). In addition, *A. baumannii* KAB02, KAB05, KAB06, AB329, and NIPH17

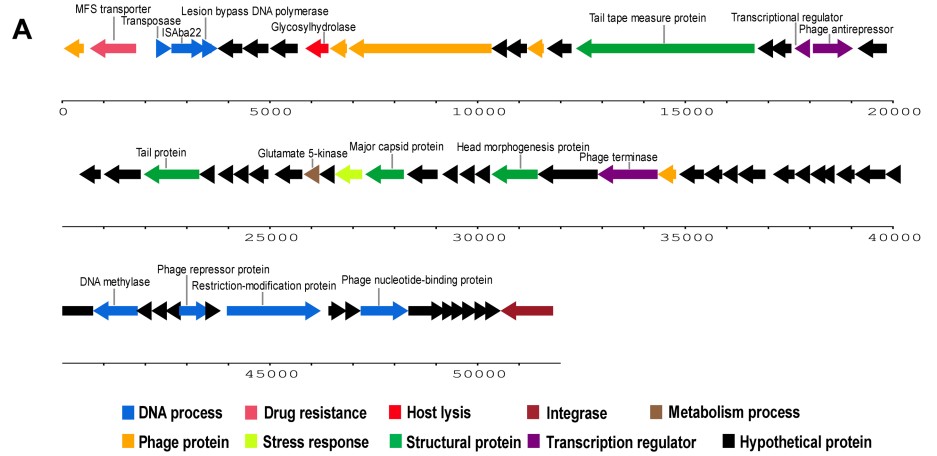

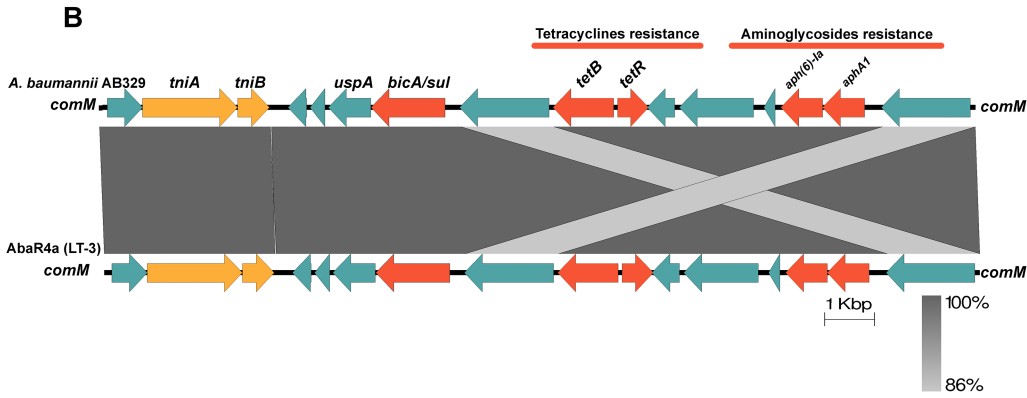

**Figure 4  Structure of intact prophage and AbaR4a identified in the *A. baumannii* strain AB329.** Structure of the intact prophage and AbaR4a identified in *A. baumannii* AB329. Genome organization of the prophage within *A. baumannii* AB329 (3A) and comparison of the genetic arrangement within *A. baumannii* AB329 with AbaR4a (LT-3) (GenBank: JN129845.1) (3B).

might have originated from *A. baumannii* XH856 isolated from China in 2010 since the genome sizes of the five *A. baumannii* strains were bigger than that of XH856 (Table S4). All ARGs were detected in the chromosome except the *aph* (3′)-VIa gene, which was found in pAB329b. These results implied that the AGRs can be rapidly transferred or passed from parent to offspring and can cause clone outbreaks in the hospital. To date, most *A. baumannii* strains display intrinsic resistance genes such as $bla_{OXA-51}$ and $bla_{ADC}$ ($bla_{AmpC}$), while acquired resistance was detected in 19% to 31% of bacterial isolates (*Kyriakidis et al., 2021*). The antibiotic susceptibility pattern of *A. baumannii* AB329 revealed high resistance to many beta-lactam antibiotics (Table S2). We detected the intrinsic resistance genes $bla_{OXA-51}$ and $bla_{ADC-25}$ as well as $bla_{OXA-23}$ and $bla_{TEM-1D}$ in the genome of *A. baumannii* AB329. However, other beta-lactamase genes that were previously reported, such as $bla_{PER-1}$, $bla_{NDM-1}$, $bla_{SPM}$, $bla_{SIM}$, $bla_{VIM}$, $bla_{GIM}$, and $bla_{IMP}$, were not detected (*Leungtongkam et al., 2018a*; *Hassan et al., 2021*; *Kongthai et al., 2021*). In addition, four

classes of efflux pumps, including the MFS, RND, SMR, and MATE families, are associated with the antimicrobial resistance of *A. baumannii* (*Abdi et al., 2020*). Consistent with a previous report, numerous *virulence* factors were detected in *A. baumannii* AB329 (*Leal et al., 2020*). Most of the virulence genes were detected in the other 292 *A. baumannii* strains isolated worldwide, and compared to the ARG patterns, the virulence gene patterns of XDRAB were not considerably different among the 292 *A. baumannii* strains (Figs. 2A and 2B). These findings indicated that all *A. baumannii* strains were derived from the same ancestor and employed the same pathogenic mechanisms to cause disease. In contrast, horizontal gene transfer of ARGs is important for the difference in ARG patterns, which leads to a critical problem in the treatment of *A. baumannii* infection.

Many of the virulence genes and ARGs are located in MGEs, such as plasmids, GIs, transposons (Tn), and prophages. These elements can move between genomes through bacterial HGT. In this study, we detected pAB329b, a novel, uncharacterized, 87-kb plasmid. The genome structure of pAB329b is a conjugative plasmid classified as the GR6 plasmid and belongs to LN_1 as well as pACICU2 (*Salgado-Camargo et al., 2020*). pAB329b might be derived from pACICU2 since it shares the same DNA backbone. *A. baumannii* strains carrying pACICU2 were isolated in 2005 in Italy, pAB329b was isolated in 2015 in Thailand, and pAB329b acquired the *aph(3′)-* VIa gene by horizontal gene transfer (Table S5).

Conjugation experiments demonstrated that amikacin resistance could be transferred from *A. baumannii* AB329 donors to recipient, sodium azide-resistant *A. baumannii* isolates. We detected the *aph* (3′)-VIa gene, plasmid GR6 and *traU* gene from the *tra* operon, which is important for generating the F-pilus in transconjugants. A previous study determined that the $bla_{OXA-23}$, $bla_{PER-1}$, and *aphA6* genes could be transferred between *A. baumannii via* the plasmid group GR6 or class 1 integrons (*Int1*) (*Leungtongkam et al., 2018b*). We were unable to find *Int1* as well as other classes (*Int2 and Int3*) in the genome of *A. baumannii* AB329, which was consistent with a study by *Ploy et al. (2000)*.

Prophages are important MGEs that encode toxins, enzymes, or drug resistance genes that allow their host to become more virulent and contribute to the evolution of pathogenic bacteria. *In silico* analysis by *Loh et al. (2020)* identified numerous ARGs that encoded beta-lactamase enzymes, N-acetyltransferases, aminoglycoside phosphotransferases, and a macrolide efflux pump in 177 prophages identified in *A. baumannii* genomes. However, we detected only the gene encoding the MFS transporter in the genome of *A. baumannii* AB329. The presence of the MFS transporter was reported in the prophage of *A. baumannii* NCIMB8209, which is involved in DNA transport and is necessary for biofilm formation (*Repizo et al., 2020*).

Accessory genes derived from HGT are found in typical regions known as GIs. Previous studies identified 63 GI loci in *A. baumannii*, and genes located within G4aby, G4abn, and G5abn were found to correspond to the resistance regions previously described as AbaR1, AbaR3, and AbaR4 (*Di Nocera et al., 2011*). AbaR4 was found in the genome of *A. baumannii* AB329. The AbaR4-type resistance island was the predominant type revealed to be a clone prevalent in most Asian countries; however, diverse variants of ARGs located within the island were found (*Kim, Park & Ko, 2012*; *Kim et al., 2013*). IS

is a short DNA sequence that plays an extensive role in bacterial adaptation to antibiotic selective pressures. A previous study on 976 *A. baumannii* genomes detected 29 IS elements (*Wright et al., 2017*). IS*Aba1* is widely distributed in *A. baumannii* and plays a major role in the transfer and expression of $bla_{OXA-23}$ and $bla_{ADC}$ (*Turton et al., 2006*; *Mugnier, Poirel & Nordmann, 2009*; *Joshi et al., 2017*). In this study, 17 IS*Aba1* genes were detected in *A. baumannii* AB329 and were found upstream/downstream of $bla_{ADC}$ (formaly $bla_{ampC}$) and $bla_{OXA-133}$. A previous report stated that the $bla_{NDM-1}$ gene was located within transposon Tn125 and was bracketed by two copies of IS*Aba125*. In this study, the $bla_{NDM-1}$ gene was found to be absent in *A. baumannii* AB329; instead, IS*Aba125* was observed to be located upstream and downstream of *aph(3′)-VIa* in pAB329b.

## CONCLUSIONS

In this study, we presented a whole-genome analysis of *A. baumannii* AB329, an XDRAB strain isolated from Thailand. The *A. baumannii* AB329 genome contained MGEs, such as two plasmids, one intact prophage, 34 IS elements, and 19 GIs. Most ARGs were located in MGEs, suggesting that these MGEs function as major mechanisms for the dissemination of ARGs in *A. baumannii*.

## ACKNOWLEDGEMENTS

We are grateful to Jason Sahl, Ph.D. and Watchanan Chantapakul for performing the LS-BSR installation.

### Funding

This work was supported by National Research Council of Thailand (NRCT) under the Mid-career Researcher grant (RSA6180042). Udomluk Leungtongkam was supported by The Royal Golden Jubilee Ph.D. Program (PHD/0227/2560). The funders had no role in study design, data collection and analysis, decision to publish, or preparation of the manuscript.

### Grant Disclosures

The following grant information was disclosed by the authors:
National Research Council of Thailand (NRCT) under Mid-career Researcher grant: RSA6180042.
The Royal Golden Jubilee Ph.D. Program: PHD/0227/2560.

### Competing Interests

The authors declare there are no competing interests.

### Author Contributions

- Supat Khongfak performed the experiments, analyzed the data, prepared figures and/or tables, authored or reviewed drafts of the article, and approved the final draft.

- Rapee Thummeepak performed the experiments, analyzed the data, prepared figures and/or tables, authored or reviewed drafts of the article, and approved the final draft.
- Udomluk Leungtongkam performed the experiments, analyzed the data, prepared figures and/or tables, authored or reviewed drafts of the article, and approved the final draft.
- Kannipa Tasanapak conceived and designed the experiments, authored or reviewed drafts of the article, and approved the final draft.
- Aunchalee Thanwisai conceived and designed the experiments, authored or reviewed drafts of the article, and approved the final draft.
- Sutthirat Sitthisak conceived and designed the experiments, analyzed the data, prepared figures and/or tables, authored or reviewed drafts of the article, and approved the final draft.

### Ethics

The following information was supplied relating to ethical approvals (i.e., approving body and any reference numbers):

The protocol was approved by Naresuan University Institutional Biosafety Committee.

### DNA Deposition

The following information was supplied regarding the deposition of DNA sequences:

The complete genome is available at NCBI GenBank: CP091452 (chromosome), CP091453 (pAB329a), and CP091454 (pAB329b).

### Data Availability

All raw data are available in the Supplementary File.

### Supplemental Information

Supplemental information for this article can be found online at http://dx.doi.org/10.7717/peerj.13718#supplemental-information.

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
