# Peer review of "Insights into mobile genetic elements and the role of conjugative plasmid in transferring aminoglycoside resistance in extensively drug-resistant Acinetobacter baumannii AB329"

_PeerJ, doi:10.7717/peerj.13718_

## Round 0.1 · original submission · Minor Revisions

After reading your manuscript and the reviewer´s comments we all think it only needs minor amendments to be suitable for publication. Please respond to their comments point by point.

·

Basic reporting

Authors conducted a well designed study, well controlled and this study gives molecular information about the behavior of Acinetobacter baumannii which is one of the most important microorganisms due its ability to adquire or to develop resistance to several antibiotics
However some annotations must be done such as to include MICs of antibiotics and to clarify some points in the document.

Experimental design

Experimental desing is well conducted and disegned, just some momments.
In Supplementary material more than to include resistance interpretation, MICs must be included and to remove some antibiotics. Besides, to mention wich document was used to define breakpoint.
A. baumannii has intrisic resistance Ceftriaxone, cefotaxime, tetracycline, therefore those dont represent clinical information o microbiological decisions (to review EUCAST intrisic resistance and CLSI M100)
Line 146: to change Bertini by Bertani.
Transconjugated selection is no clear, quote- Transconjugants were selected on LB plates containing the following components: 250 μg/ml sodium azide; 50 μg/ml ticarcillin or 250 μg/ml sodium azide; 20 μg/ml kanamycin. Plates were prepared with (1) 250 sodium azide, (2) ticarcillin and (3) or sodium azide (again). To clarify this point.
In table 2, to include MICs of Ak resistance after transconjugation. Besides to remove those antibiotics that are intrinsically resistant

Validity of the findings

No comment

Additional comments

Line 32: In silico must be written in italics.
Line 55: after bla OXA-23 in subscript
Line 58: after bla, GES-11 in subscript
OXA-23 was the most common gene present in several mobile genetic elements such as ISAb4, ISAb10, Tn2006, Tn2007 and Tn2008 inorder to have a better imagen of how common was this element to change redaction, Suggestion: blaOXA-23 was found in..... in this way is easier to follow the findings
Line 56: ISAba4 and ISAb10, after nomenclature of insetion sequence (IS) Aba in italics.
Line 297: In silico in italics
Some elements that confers intrisic resistance such as OXA-51 and ADC (a AmpC) must be included in discussion.

Reviewer 2 ·

Basic reporting

The english should be revised.
The quality of the figures should be improved.

Experimental design

No comment

Validity of the findings

No comment

Additional comments

The manuscript “Insights into mobile genetic elements and the role of conjugative plasmid in transferring aminoglycoside resistance in extensively drug-resistant Acinetobacter baumannii AB329” describe the genomic analysis of an A. baumannii strain causing an outbreak in Thailand. The analysis reports the identification of two plasmids, one of them a conjugative megaplasmid, and several prophages, genomic islands and insertion sequences. They conclude that all these MGEs are responsible for the acquisition and transference of antimicrobial resistance in this strain.
The work is well elaborated and offer a general vision of the role of all these MGEs in the antimicrobial resistance acquisition.
Minor comments:
1- Line 36: the authors should include the definition of conjugative plasmid, as this is an important part of the work.
2- Line 40: which is the relevance of the plasmid GR6? Is this included in one of the 21 lineages? Please, give more information about this plasmid.
3- Line 43: define prophage.
4- Line 46: as this sentence is writen it seems that the 177 strains were analysed in this work. Please write it in a way that it is well understood.
5- Line 48: include the definition of genomic island.
6- Line 53: include the definition of insertion sequence.
7- Line 62: the authors should include more information about this outbreak.
8- Line 133: the authors should indicate the tool employed to identify the genomic island.
9- Line 152: the first section of the results would be better structured if it were divided into two parts, one including the antibiotic susceptibility and other with the genome sequence.
10- Line 160: the authors should include the genome size of the plasmids.
11- Line 161: The number of ORFs and tRNA are the total of the chromosome or include the plasmids? The authors should explain it.
12- Line 178: which is the difference between shell genes and cloud genes?
13- Line 251-260: the authors should base this hypothesis in previous published works. Include reference that reinforce your hypothesis.
14- Figure 1b: this figure is elongated. Please, improve it.
15- Figure 2: in this figure is very difficult to read the gene names.
16- Figure 4. Improve the figure quality.
Major comments
In the section of Conjugative transfer of aminoglycoside resistance gene of plasmid pAB329b, the authors determine the transference of the aminoglycoside resistance by conjugation, but the don´t explain if this transference is only of the gene or the complete plasmid. Besides, if the transference only implies the gene, is this due to the effect of a transposase?

Reviewer 3 ·

Basic reporting

The manuscript was clear described. and it also contain enough information and backgroud. and its structure was accptable. also the raw data was shared.

Experimental design

The research question was well designed. and the methods here is ok to replicate.

Validity of the findings

whole genome sequencing and othere experiment was performed. and the result support their hypothesis.

Additional comments

Line 57: Tn2009(OXA-23) was missing
Line 81: the strain used in this study is quite old. it would be difficult to represent the current situation.
Line 82: did AST followed CLSI? which version?
Line 130: which paster version server was used in this study? 2?
Line 163: how about the allele variant of ST86 to ST2? it should be stated in the manuscript.
Line 215: the detail of the loss of aph3-VIa should be analysis.
The easyfig figure should be improved. e. g. the blastn length should be set.

---

## Round 0.2 · accepted · Accept

Based in the recommendations of 2 reviewers I am glad to tell you that your paper is ready for publication in PeerJ.

·

Basic reporting

Authors have made all changes according suggestions. Work is clearer and fluid.

Experimental design

Authors have made all changes according suggestions in methodology. Work now is clearer and fluid.

Validity of the findings

No comment

Additional comments

No comment

Reviewer 3 ·

Basic reporting

The questions were all addressed.

Experimental design

The questions were all addressed.

Validity of the findings

The questions were all addressed.

Additional comments

The questions were all addressed.